# Neurotrophins and Their Receptors: BDNF’s Role in GABAergic Neurodevelopment and Disease

**DOI:** 10.3390/ijms25158312

**Published:** 2024-07-30

**Authors:** Carlos Hernández-del Caño, Natalia Varela-Andrés, Alejandro Cebrián-León, Rubén Deogracias

**Affiliations:** 1Instituto de Neurociencias de Castilla y León (INCyL), 37007 Salamanca, Spain; chernandez@usal.es (C.H.-d.C.); nvarela@usal.es (N.V.-A.); alexcele@usal.es (A.C.-L.); 2Instituto de Investigación Biomédica de Salamanca (IBSAL), 37007 Salamanca, Spain; 3Departamento de Biología Celular y Patología, Facultad de Medicina, Universidad de Salamanca, 37007 Salamanca, Spain

**Keywords:** neurotrophins, BDNF, TrkB, GABAergic neurons, neurodevelopment, autism, schizophrenia, Rett syndrome

## Abstract

Neurotrophins and their receptors are distinctly expressed during brain development and play crucial roles in the formation, survival, and function of neurons in the nervous system. Among these molecules, brain-derived neurotrophic factor (BDNF) has garnered significant attention due to its involvement in regulating GABAergic system development and function. In this review, we summarize and compare the expression patterns and roles of neurotrophins and their receptors in both the developing and adult brains of rodents, macaques, and humans. Then, we focus on the implications of BDNF in the development and function of GABAergic neurons from the cortex and the striatum, as both the presence of BDNF single nucleotide polymorphisms and disruptions in BDNF levels alter the excitatory/inhibitory balance in the brain. This imbalance has different implications in the pathogenesis of neurodevelopmental diseases like autism spectrum disorder (ASD), Rett syndrome (RTT), and schizophrenia (SCZ). Altogether, evidence shows that neurotrophins, especially BDNF, are essential for the development, maintenance, and function of the brain, and disruptions in their expression or signaling are common mechanisms in the pathophysiology of brain diseases.

## 1. Historical Perspective on Neurotrophins

Neurotrophins are a family of secreted proteins consisting of the nerve growth factor (NGF), the brain-derived neurotrophic factor (BDNF), neurotrophin-3 (NT-3), and neurotrophin-4/5 (NT-4/5). 

Building upon the initial work of Spemann and Mangold regarding the “embryonic organizer” and embryonic development [1], the works of Viktor Hamburger and Rita Levi-Montalcini between the 1940s and the early 1950s led to the identification of NGF as the first factor promoting the survival and differentiation of sensory and sympathetic neurons [2,3,4].

In the early 1980s, the works by Yves-Alain Barde and Hans Thoenen led to the purification from pig brain of a factor that supported survival and growth of embryonic chick sensory neurons, later identified as BDNF [5]. The subsequent research revealed that BDNF plays a crucial role in promoting the survival and growth of neurons in the central nervous system and that BDNF disruption is involved in various neurological and psychiatric disorders [6]. These studies suggested that BDNF may play a crucial role in maintaining healthy brain functions and preventing the onset of mental illnesses, sparking a wave of research aimed at developing novel therapeutic interventions that target BDNF to treat these conditions, such as the development of therapies that aim to harness the power of BDNF to repair damaged neurons and restore brain functions [7]. 

In the early 1990s, a series of experiments that sought to isolate and characterize novel molecules analog to NGF and BDNF led to the discovery of NT-3 [8,9,10,11] and NT-4/5 [12,13,14,15].

In this review, we summarize the main characteristics and functions of the neurotrophins and their receptors. We also discuss the role of BDNF in the brain GABAergic inhibitory neurons, with special focus on two brain areas: the cerebral cortex and the striatum. Finally, we recapitulate the effects of BDNF signaling disruption in GABAergic neurons and its relationship with neurodevelopmental disorders.

## 2. Neurotrophins Expression, Synthesis, Structure, and Secretion in Brain

### 2.1. Expression in the Developing Brain

During brain development, neurotrophins exhibit distinct spatial and temporal patterns of RNA expression and protein distribution (Figure 1). In the embryonic rat brain, the most abundant neurotrophin transcripts are *Nt-3* and *Nt-4/5*, while *Ngf* expression is intermediate, and *Bdnf* is scarcely detected. These patterns shift after birth. In postnatal development, while *Ngf* RNA levels are relatively constant in the entire brain, *Nt-3* expression decreases while *Bdnf* expression increases until adulthood. *Nt-4/5* RNA levels decay to intermediate levels and then remain stable (Figure 1A) [10,15].

In rodents, it has been observed that the expression levels of different neurotrophins vary not only through developmental stages but also between brain regions (Figure 1B). For instance, in newborn rats, *Nt-3* expression is higher in the cerebellum, hippocampus, and cortex, which are more immature areas at this stage, whereas in the adult it remains high only in the hippocampus [10]. Conversely, *Bdnf* expression in the newborn is primarily located in areas that mature earlier, such as the hindbrain, midbrain, and diencephalon, as well as in the hippocampus. In the adult brain, the highest levels of *Bdnf* RNA are found in the hippocampus, with abundant levels in the neocortex, diencephalon, midbrain, and cerebellum and decreasing levels in the hindbrain and spinal cord [10,15,16,17]. It is noteworthy that neither *Nt-3* nor *Bdnf* are expressed in the striatum during both the newborn and adult stages [10]. Finally, *Nt-4/5* displays a similar expression pattern in different brain regions along postnatal development, except in the cerebellum, where it shows several expression peaks [15].

**Figure 1 ijms-25-08312-f001:**
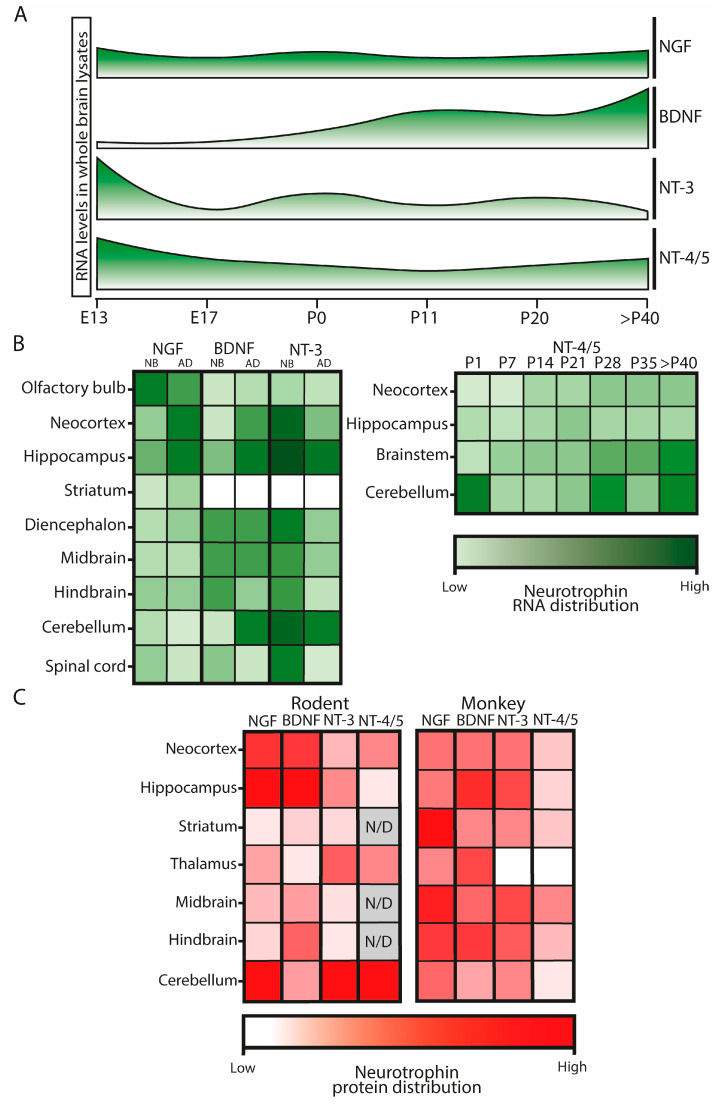
Schematic representations of neurotrophin RNA and protein relative level distribution change along brain development and comparing between species. (**A**): Neurotrophin RNA levels in whole brain lysates along rat development. Data acquired from northern blot assays from reference [10] and RNAse protection assays from reference [15] using probes against the coding sequence (CDS) of each neurotrophin. E: embryonic gestation day; P: postnatal day. (**B**): *Ngf, Bdnf*, and *Nt-3* RNA levels in different brain regions of newborn (NB) and adult (AD) rats, and *Nt-4/5* along postnatal rat development in different brain regions. Data acquired from northern blot assays from reference [10] and RNase protection assays from reference [15], using probes against the CDS of each neurotrophin. (**C**): NGF, BDNF, NT-3, and NT-4/5 protein distribution in different brain areas from adult rodents and monkeys. Rodent data were acquired from both immunohistochemistry and protein quantification experiments from references [18,19,20,21]. Monkey data were obtained from immunohistochemistry experiments from reference [22]. In all cases, used antibodies do not discriminate between precursors and mature neurotrophins. Darker colors mark higher RNA or protein levels, relativized to the maximum expressing point. N/D: data not available.

### 2.2. Synthesis, Secretion, and Protein Levels

Neurotrophins, like most of growth factors, are initially synthesized in the rough endoplasmic reticulum as precursors called proneurotrophins, composed of a N-terminal prodomain and a C-terminal mature domain. These proneurotrophins are then packaged into secretory vesicles for processing by convertase family proteases, leading to the production of mature neurotrophins. In the case of BDNF, it is initially synthesized and glycosylated as a precursor (pro-BDNF), which then moves into the Golgi apparatus. Here, it may undergo endoproteolytic cleavage, promoting its targeting to secretory granules, or it may be directly secreted as a proneurotrophin, which can be processed by extracellular proteases or act as an independent signal [23].

As secreted molecules, protein levels and distribution do not directly correspond to their expression pattern. Also, both the locations and the protein levels of neurotrophins differ between species (Figure 1C) [17,18,19,20,21,22]. Adult rodents show the highest immunoreactivity for NGF in the cortex, the hippocampus, and the cerebellum [19], whereas in the rhesus monkey the highest immunoreactivity for NGF is detected the striatum, the midbrain, and the hindbrain [22]. BDNF immunoreactivity is higher in the rodent neocortex, hippocampus, and hindbrain [20,21], while in monkeys the highest immunoreactivity is located in the hippocampus, hindbrain, and thalamus/hypothalamus [22]. NT-3 levels are higher in the rat hippocampus, thalamus, and cerebellum [18,21], while in the macaque they are higher in the hippocampus, midbrain, and hindbrain [22]. NT-4/5 levels are especially abundant in the cerebellum in rodents [21], whereas in the monkey they are low throughout the whole brain except in the midbrain. Notably, both NT-3 and NT-4/5 are absent in the macaque thalamus [22].

### 2.3. BDNF Expression and Protein Location during Development

During development, the expression of *BDNF* is dynamically regulated in a region-specific manner. In both mice and human the gene encoding BDNF contains several 5′-non-coding exons driven by distinct promoter regions, resulting in a common protein-coding 3′ exon-mRNA [17,24,25]. These mRNA variants create a “spatial code” with distinct subcellular distribution of these transcripts in neuronal and non-neuronal cells, implicating them in many different functions [17,26]. For example, regulation of dendrite complexity [27], thermogenesis, body weight [28], aggression in male mice [29], impaired maternal care [30], and impaired inhibitory synapse formation [31,32] are such functions. 

In mice, *Bdnf* expression increases throughout development until adulthood, with the mouse hippocampus particularly enriched in *Bdnf* transcripts during postnatal development. In humans, the highest expression peak is observed in the thalamus during infancy, with a similar progression in the amygdala, cerebellum, and dorsolateral prefrontal cortex during development. However, almost no mRNA can be detected in the human striatum at any developmental stage. The levels, proportions, and distributions of the various *BDNF* mRNA variants change differently during brain development across species (Figure 2A) and even between mouse strains [17].

BDNF protein levels also increase during postnatal development, aligning with the maturation times of different brain areas (Figure 2B). Early maturing regions such as the thalamus, hypothalamus, midbrain, and hindbrain exhibit higher BDNF levels at earlier time points during postnatal development, while other regions like the olfactory bulb, neocortex, hippocampus, and striatum begin to show increased BDNF levels from approximately P10. The cerebellum displays a later increment of BDNF levels, being one of the last brain areas to develop [17,33].

The subcellular distribution of BDNF varies among brain regions (Figure 2C). In adult rats, BDNF immunoreactivity is higher inside the cell bodies in neurons located in the neocortex and cerebellum, while in the hippocampus, striatum, thalamus, hypothalamus, midbrain, and hindbrain, it is more prominent in the fibers [20]. This distribution, akin to the subcellular distribution of the alternative mRNAs [26], also represents a “spatial code” that likely regulates BDNF functions. Whether this “code” remains constant during development or changes with age to exert different roles is still unknown.

**Figure 2 ijms-25-08312-f002:**
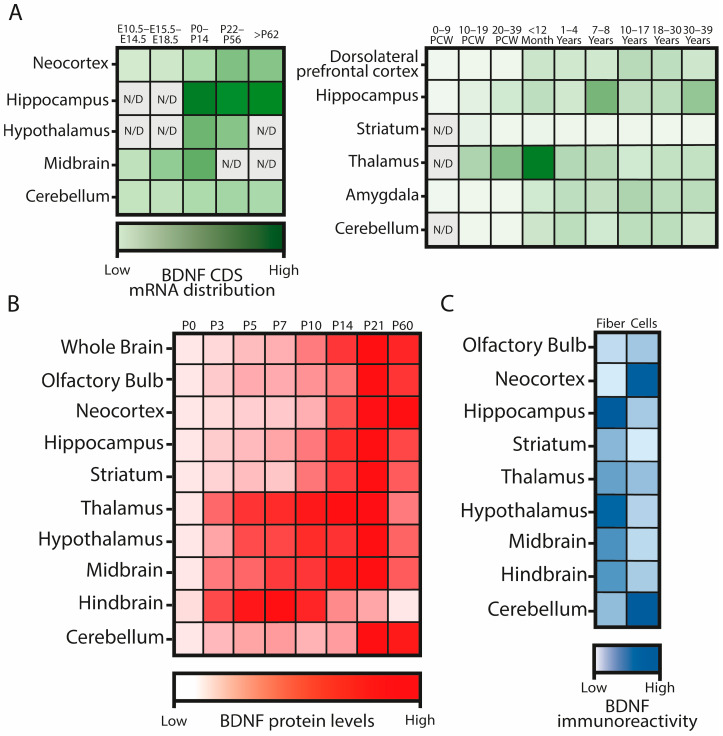
Schematic representations of BDNF mRNA and protein distribution in the developing brain. (**A**): *BDNF* CDS-mRNA distribution in different brain areas in mouse (left) and human (right) along development, according to RNAseq data from reference [17]. Darker colors mark higher mRNA levels, relativized to the maximum expressing point. (**B**): BDNF protein levels along mouse postnatal development in different brain areas, considering the mean of the detected protein levels by western blot in both BALB/C and C57BL6/J mice in reference [17]. (**C**): BDNF subcellular distribution in different brain areas in the adult rat brain, according to reference [20]. E: mouse embryonic stage; P: mouse postnatal day; PCW: human postcoital week. N/D: data not available.

## 3. Neurotrophin Receptors

Neurotrophins exert their effects by binding with high affinity to specific cell surface receptors known as tropomyosin receptor kinases (Trks), namely TrkA, TrkB, and TrkC. These tyrosine kinase receptors dimerize upon ligand binding and undergo transphosphorylation, where the intracellular kinase domain of the receptor phosphorylates itself and promotes the activation of intracellular protein effectors [34,35]. Alternatively, the non-processed neurotrophins or proneurotrophins, as well as the mature fully processed neurotrophins, bind to the low-affinity p75 neurotrophin receptor (p75^NTR^) [36,37,38,39,40]. 

### 3.1. Trk Receptors

Trk receptors are expressed differentially during development (Figure 3B) [41] and in different subsets of neurons, playing a wide range of roles after neurotrophin binding. Surprisingly, Trk receptors can also be activated in the absence of neurotrophins by trans-activation mechanisms [42,43,44,45,46,47,48]. Therefore, it is not surprising that some phenotypes observed in Trk-deficient mice [49] do not totally phenocopy the ones observed in neurotrophin-deficient mice [50,51,52,53].

The structure of various Trk receptors is very similar (Figure 3A): an extracellular domain composed of one cysteine-rich domain followed by three leucine-rich motifs and another cysteine-rich domain, two immunoglobulin-like C2 domains, a transmembrane domain (which is responsible for dimerization after neurotrophin binding), and a cytoplasmic tyrosine kinase domain that guides the downstream signaling. However, the existence of several splicing isoforms of all Trk receptors that differ in the cytoplasmic domain has been reported [34]. 

Upon neurotrophin binding, Trk receptors activate three different signaling pathways, the PLCγ/IP3, the PI3K/Akt, and the MAPK pathways, which mediate multiple and specific neurotrophin-induced biological responses [54]. 

In general, these responses are modulation of gene expression, presynaptic neurotransmitter release, postsynaptic neurotransmitter receptor exposure and function, ion channel activity and conductance, regulation of dendritic growth and morphology, and spine maturation [55].

In addition to their canonical direct activation, Trk receptors can be transactivated through alternative neurotrophin-independent pathways [44]. The transactivation of Trk receptors was initially observed in studies investigating the signaling pathways of the adenosine and pituitary adenylate cyclase-activating polypeptide (PACAP) through G-protein-coupled receptors [42,43]. These studies provided early insights into the ability of Trk receptors to be activated through non-canonical pathways. More recently, in vivo studies have demonstrated the transactivation of TrkB and TrkC by the epidermal growth factor (EGF) to drive the migration of newborn mouse cortical neurons [47]. Additionally, transactivation of TrkB has been observed in response to glucocorticoids [45], zinc [46], and, more recently, to oxytocin [48].

Among the different Trk isoforms (Figure 3A), the most well studied is TrkB-T1, a truncated TrkB isoform lacking the intracellular tyrosine kinase domain but with a small aminoacidic tail that transduces BDNF signaling into the cytosol [34,56]. TrkB-T1 presents a unique cytoplasmic tail of 11 amino acids fully conserved across species, suggesting a potential important evolutionary biological role [57]. TrkB-T1 is mainly, but not exclusively, expressed in astrocytes [58], and the function of its cytoplasmic tail is still not fully understood. However, recent research has implicated TrkB-T1 in various biological processes, including neural development, early embryonic central nervous system and mesenchymal development, glioma biology, neuropathic pain, regulation of cell morphology, and the functionality of glycine transporters [58,59]. In addition, TrkB-T1 has been shown to exhibit an important neuroprotectant function [60,61], and it has been reported to interact with p75^NTR^ in modulating functional and structural plasticity in hippocampal neurons [62].

### 3.2. The p75 Neurotrophin Receptor

The receptor p75^NTR^ plays a significant role in binding NGF, BDNF [36], NT-3 [37], and NT-4/5 [38]. Structurally, it consists of an extracellular domain with four cysteine-rich subdomains that binds the four mature neurotrophins with similar affinity. Its intracellular portion contains a chopper domain and death domain (Figure 4) [40]. 

During the development of the central and peripheral nervous system, p75^NTR^ expression is high, particularly during synaptogenesis and developmental cell death. However, its expression is downregulated in adulthood (Figure 5C) [17]. Nonetheless, following injury or disease, p75^NTR^ expression can increase once again [63]. 

It is worth noting that proneurotrophins can be directly secreted in disease states and interact with p75^NTR^ [23]. Increased levels of pro-NGF have been observed in the frontal and parietal cortex of Alzheimer’s patients compared to controls [64,65]. Conversely, in a cleavage-resistant pro-BDNF knockin mouse, secretion of pro-BDNF has been shown to negatively regulate synapse plasticity and transmission [66]. 

The interaction between proneurotrophins and p75^NTR^ can induce apoptosis, even at sub-nanomolar concentrations. This occurs through the formation of a heterodimer between p75^NTR^ and the type I transmembrane protein sortilin. Sortilin recognizes the proneurotrophin prodomain, while p75^NTR^ recognizes the domain corresponding to the mature neurotrophin. This apoptotic mechanism involves the activation of the JNK-p53 apoptotic pathway and subsequent procaspase-mediated cleavage of different substrates [67,68,69,70]. Furthermore, p75^NTR^ has been associated with cytoskeletal reorganization through interactions with proteins such as RhoA [71] and fascin [72]. p75^NTR^ can also be cleaved by the α- and γ-secretases, splitting it into its extracellular (p75^ECD^) and intracellular (p75^ICD^) domains. p75^NTR^ then interacts through its chopper domain with the intracellular domain of TrkA receptors establishing a tetramer (Figure 4). This interaction allosterically modulates TrkA binding affinity for NGF, increasing it and mediating effects on cell survival and differentiation [40]. It also plays a role in neuronal morphology, including dendritic arborization, axonal pruning, and neurite retraction [73,74,75]. Neurotrophins, including both proneurotrophins and mature forms, have specific effects on synaptic plasticity. The secretion of pro-BDNF and mature BDNF is influenced by different neuronal activities, with pro-BDNF being predominant in low-frequency stimulation associated with long-term depression (LTD), and mature BDNF being released following high-frequency stimulation associated with long-term potentiation (LTP) [76,77]. In the CA1 area of the hippocampus, pro-BDNF enhances LTD, while mature BDNF is required for the maintenance of LTP [55,78]. Moreover, neurotrophins and p75^NTR^ are involved in the control of cell cycle and proliferation. Neurons are typically in a quiescent state, but cell cycle reactivation can lead to apoptosis or transition to tetraploid neurons. Activation of cell proliferation factors or neurotrophic factor deprivation can trigger this process, and in disease states like Alzheimer’s, proNGF activation ultimately results in neuronal death [79,80].

## 4. Role of BDNF in GABAergic Neurons

BDNF plays distinct roles in glutamatergic and GABAergic neurons. The BDNF/TrkB signaling pathway in the hippocampus is critical for proper synaptic function, and disruptions in this pathway have been implicated in various diseases [81]. The dysregulation of BDNF signaling can have significant consequences for synaptic plasticity and overall hippocampal function. In the hippocampus, BDNF is involved in mediating the formation of functional synapses in both types of neurons [82]. However, in hippocampal GABAergic populations, BDNF not only influences synapse formation but also modulates their growth, differentiation, and synaptic plasticity [53,82,83,84]. Interestingly, in hippocampal glutamatergic neurons, BDNF does not control neuronal growth but rather the neurotrophin NT-3 takes on that role [82]. This highlights the multifaceted role of BDNF in shaping the development and function of GABAergic neurons in the hippocampus and demonstrates the specificity of neurotrophin regulation in different neuronal populations within the same brain region. 

Furthermore, BDNF also plays a role in specifically regulating GABAergic populations in other brain areas such as the cortex or the striatum. 

### 4.1. Role of BDNF in the GABAergic Cortex

Neurogenesis in rodents begins around embryonic stage E9.5 with the division of neuroepithelial cells, leading to the formation of radial glial cells and basal progenitors [85]. These precursor cells then undergo radial migration to form the cortical layers, with deeper layers migrating first and outer layers arriving last. 

During this process, BDNF and NT-3 expression in the brain is minimal [10]. However, the signaling of TrkB and TrkC receptors, transactivated by the epidermal growth factor receptor (EGFR), regulates this migration [47]. The addition of exogenous BDNF during these embryonic stages leads to the induction of neuronal bone morphogenetic protein 7 (BMP7) expression through the BDNF/MAPK/ERK1/2 pathway, resulting in premature and incorrect differentiation of radial glial cells and impaired neuronal migration [86]. 

The integration of interneurons in the cortex occurs primarily during early postnatal development and involves programmed cell death and circuit refinement [87,88]. BDNF levels begin to rise at this point [10], and BDNF is known to play a crucial role in the maturation of GABAergic interneurons. Specifically, BDNF is required for inhibitory cortical neurons’ dendritic development but not for that of excitatory neurons [89]. This effect on inhibitory neurons is mainly seen during the late period of corticogenesis, where BDNF supports the maintenance of neuron size and dendrite structure [90]. 

Among the interneurons, parvalbumin-expressing neurons appear to be particularly dependent on BDNF and TrkB signaling, which regulates parvalbumin expression during early postnatal development in the visual and prefrontal cortex [91,92]. GABA transmission in cortical interneurons is modulated by BDNF, which influences the transcription of the GABA-synthesizing enzyme GAD65 in a MAPK/ERK/CREB-dependent manner [93]. On the other hand, it has been observed that hyperactivation of the MEK1/ERK1/2 signaling pathway leads to specific defects in cortical parvalbumin neurons, resulting in impaired behavioral inhibition [94]. TrkB signaling also regulates the dynamics of parvalbumin cortical neurons, modulating their excitability and the number and strength of synapses they make with pyramidal neurons [95]. TrkB, activated by both BDNF and NT-4, also mediates the expression of Kv3.1b and Kv3.2, two essential potassium channels for the activity of fast-spiking basket and chandelier cells [96].

The incoming synapses onto parvalbumin neurons are heavily regulated by the perineuronal net (PNN), which influences their function [97]. The development of the PNN depends on BDNF and JNK signaling, and disruptions in the PNN have been linked to schizophrenia [98].

p75^NTR^ signaling also plays a significant role in the development and function of parvalbumin neurons, as its expression is observed in early cortical progenitors and persists in post-mitotic cortical neurons and migrating neurons within the cortical plate [99]. Conditional deletion of p75^NTR^ in mice results in impairments in the development of cortical interneurons and upper-layer pyramidal neurons, leading to the loss of cortical layer thickness and a decrease in the number of parvalbumin, somatostatin, neuropeptide Y, and calretinin-positive cells [99]. p75^NTR^ is also required for the survival of cortical neuron progenitors and the production of later-born neurons. In parvalbumin neurons, p75^NTR^ regulates the timing of maturation and connectivity in the visual cortex, affecting the development of their perineuronal nets and dynamically regulating them through pro-BDNF-dependent plasticity mechanisms [100]. Deletion of p75^NTR^ causes perineuronal net aggregation in the prefrontal cortex, which can be rescued by reintroducing p75^NTR^ in preadolescent, but not postadolescent, prefrontal parvalbumin neurons [101]. These results show that p75^NTR^ signaling not only is relevant in disease states but is also essential for proper cortical wiring, indicating as well that more research needs to be conducted regarding p75^NTR^ physiological function in the normal development of the cortex and other brain areas.

### 4.2. Role of BDNF in the Striatum

The striatum, a prominent GABAergic region within the basal ganglia, is primarily composed of Medium Spiny Neurons (MSNs) derived from the lateral ganglionic eminence, along with several interneuron populations from various sources [102]. 

The MSNs receive glutamatergic inputs from cortical and thalamic areas, as well as dopaminergic inputs from the substantia nigra pars compacta and the ventral tegmental area [103]. The MSNs also undergo local modulation by interneurons and other MSNs [104].

There are functionally distinct populations of MSNs, expressing either the dopamine receptor D1 or D2 [105]. The MSNs expressing the D1 receptor project onto the substantia nigra, while the D2 project into the globus pallidus, forming the direct and indirect dopaminergic pathways, respectively [104,106].

Functionally, the striatum is mainly involved in the cognitive planning of purposive motor acts [107], but it also participates in other functions like decision-making, reward processing, and response inhibition, among others [108].

In the adult striatum, while *Bdnf* mRNA is absent [10], BDNF protein levels are high as it is anterogradely transported to the striatum from the cerebral cortex, substantia nigra, amygdala, and thalamus [17,20,109]. BDNF secretion in the corticostriatal circuitry exhibits a specific distribution pattern. Through the utilization of retrograde viral tracing, it has been observed that neurons expressing *Bdnf* originating from the medial prefrontal cortex primarily project onto the dorsomedial striatum. Conversely, neurons originating from the motor and the agranular insular cortices extend their axons to the dorsolateral striatum. Moreover, the outputs from the orbitofrontal cortical neurons are directed toward the dorsal striatum, with the specific target location being dependent on the mediolateral and rostrocaudal positioning of these neurons. Notably, distinct populations of cortical neurons have been identified to release BDNF at their axon terminals, thereby influencing BDNF/TrkB signaling within different areas of the striatum [110]. This corticostriatal parcellation, in conjunction with striatal outputs, also plays a crucial role in establishing the functional subdivisions of the striatum into limbic, associative, and sensorimotor areas [111].

Furthermore, the presence of TrkB in the membrane of the MSNs is intrinsically modulated. Activation of the dopamine indirect pathway causes the retraction of TrkB from the plasma membrane [112], whereas the direct pathway enhances TrkB’s sensitivity to BDNF by modulating TrkB presence on the cell surface [113].

Conditional BDNF knockout mice have revealed the importance of BDNF in proper striatal outgrowth, MSN arborization, and functional development. Additionally, BDNF/TrkB signaling controls the number of newborn striatal neurons and supports the survival of immature MSNs [53,114,115,116].

BDNF downstream signaling also regulates striatal synaptic dynamics, influencing synapse-related genes, drug response, and addiction processes [117]. In vivo, BDNF-mediated dopamine potentiation in the striatum requires specific pathway activations, such as PI3K and MAPK, but not PLCγ [118]. However, PLCγ1 plays a crucial role in establishing the dopaminergic system and regulates dopamine release input into the striatum [119,120]. PLCγ hydrolyses PIP2 into DAG and IP3, both of which heavily modulate synaptic transmission in striatal neurons, affecting MSNs’ output neuron activity and voltage-gated sodium channels within MSNs [121,122].

Regarding TrkB truncated isoforms, TrkB-T1 is not expressed in the striatum [123] but modulates corticostriatal transmission, inhibiting BDNF signaling by NT-4/5 [124]. In vitro, excitotoxic stimulation downregulates TrkB-FL via calpains and upregulates truncated TrkB protein levels through activation of its transcription and translation [125]. ProBDNF/p75^NTR^ signaling is also essential in the striatum, mediating the survival of neuronal progenitors and the development of the basal ganglia. Disruption of p75^NTR^ has been linked with Huntington’s disease, as it affects corticostriatal synaptic LTP [126] and the survival of striatal neurons [127]. p75^NTR^ alterations are also related with Parkinson’s disease, alcohol abuse, cocaine sensitization, ischemic trauma, and HIV-associated neurological disorder [99,128,129]. However, the underlying downstream molecular mechanisms guiding these roles remain to be fully elucidated.

## 5. BDNF on Neurodevelopmental Disorders with Impairments in GABAergic Neurons

Dysfunctions in the formation of the GABAergic system are linked to the appearance of neurodevelopmental disorders, mainly due to a loss in the balance between excitation and inhibition in the brain [130]. Autism spectrum disorder (ASD), Rett syndrome (RTT), and schizophrenia (SCZ) are three neurodevelopmental disorders that show this imbalance derived from impairments within the GABAergic brain network and have been related with BDNF-signaling disruptions.

### 5.1. BDNF in Autism Spectrum Disorder

Autism spectrum disorder (ASD) is characterized by impairments in social interaction, communication, and repetitive behaviors, often accompanied by co-occurring symptoms such as dyskinesia, speech delay, sleep disorder, anxiety, and epilepsy in children and depressive symptoms in adolescents and adults [131,132]. The etiology of ASD is complex, involving abnormal interactions among genetic and environmental factors, leading to epigenetic changes that modify gene transcription and cell function [131]. 

Neurodevelopmental alterations associated with ASD involve changes in molecular processes, including chromatin remodeling and Ca^2+^ and Wnt signaling [131]. Wnt signaling, critical for cell fate determination and neural development, plays a key role in normal brain growth, neuron proliferation, and connectivity [133,134]. These alterations modify BDNF expression and signaling in the brain, as BDNF expression is regulated by neural activity and Ca^2+^ signaling via transcription factors such as CREB, NFAT, and MEF2, which are dysregulated in ASD [131,135,136].

In animal models with autism-like behavior, such as BTBR mice and neuroligin 3 (NLG3) loss-of-function mutants, impairments in GABAergic hippocampal synaptic function have been found and directly related to BDNF/TrkB signaling impairments [137,138]. 

Epigenetic mechanisms, including DNA methylation and histone modification, are also involved in ASD, affecting social behavior regulation and GABAergic transmission [131,139]. Enhanced binding of MeCP2 to *Gad1* and *Gad2* promoter regions, reducing the expression of GABA synthesizing enzymes, has been observed in ASD patients, impairing GABAergic transmission [140]. Altered expression of the oxytocin receptor, which can transactivate TrkB [48], has also been linked to ASD, showing high levels of methylation of its promoter [131]. 

Additionally, the BDNF Val66Met mutation, which diminishes activity-dependent BDNF signaling [141,142], has been associated with autism-like social deficits in mice, with differences between males and females [143]. 

The relationship between BDNF alterations and ASD is not entirely clear, as it may involve both causative and consequential factors. Neurodevelopmental alterations prior to BDNF’s functional roles in neuron maturation may lead to BDNF misregulation as a consequence of the disorder, potentially driving the appearance of various co-occurring symptoms. Conversely, disruptions in BDNF signaling can contribute to some of the behavioral deficits in ASD, serving as one of the causative factors of the disease, particularly through its role in GABAergic neuron maturation, given the strong association between GABAergic dysfunction and the appearance of ASD traits [131,140]. The intricate interplay between BDNF signaling and ASD underscores the multifaceted nature of the disorder, involving various genetic, epigenetic, and molecular mechanisms, which, to this day, are not fully understood.

### 5.2. BDNF in Rett Syndrome

Rett syndrome (RTT) is a severe neurodevelopmental disorder that primarily affects females, with rare cases affecting males. In males, RTT leads to rapid fatal neonatal encephalopathy, while in females, it involves developmental regression, including the loss of acquired language and social skills, motor impairments, autonomic dysfunction, and seizures [144,145,146].

Classical RTT is caused by loss-of-function mutations in the Methyl-CpG Binding protein 2 (MeCP2), while atypical RTT can result from mutations in other genes such as CDKL5 and FOXG1 [147].

MeCP2 is an epigenetic regulator that plays a crucial role in reading DNA methylation to mediate DNA binding of the repressors NCOR1 [148,149] and Sin3A [150,151] or of the activator transcription factor CREB [150]. 

Through these interactions, MeCP2 modulates *BDNF* expression [145], with BDNF also regulating MeCP2 phosphorylation. BDNF signaling in cortical neurons mediates several MeCP2 phosphorylations [152], influencing cortical dendritic branching, activity, and sustained effects of antidepressants [148,153,154,155].

Dysregulation of the brain’s excitatory/inhibitory balance, mainly caused by impairments in the brain’s GABAergic network, is a physiological hallmark of RTT etiology [156]. MeCP2 deletion specifically from GABAergic neurons replicates most of the phenotypical features of MeCP2 full KO mice, while restoration of MeCP2 only in GABAergic neurons rescues ataxia, apraxia, and social abnormalities, and extends lifespan [146].

Given the importance of GABAergic network impairments in RTT etiology and the molecular relationship between BDNF and MeCP2, BDNF has become a target for RTT therapies. BDNF overexpression in postnatal excitatory forebrain and hippocampal neurons has shown promise in reverting some RTT characteristics, highlighting the neuroprotective function of BDNF [157,158].

Pharmacological approaches that enhance BDNF-dependent signaling have shown promise as therapy for RTT. TrkB agonists like LM22A-4 [159] and 7,8-DHF [160] have been able to revert some RTT symptoms in animal models, including restoration of hippocampal synaptic plasticity and object location memory [161]. Additionally, Fingolimod, a compound that stimulates the MAPK/ERK signaling pathway through modulation of Sphingosine-1 phosphate receptors, has shown promise in improving RTT-like features in MeCP2 KO mice [162], although it failed to provide clear results in clinical trials [163]. Recently, the FDA approved the first Rett syndrome treatment, Trofinetide, an endogenous tripeptide of the N-terminal domain of IGF-1 [164].

### 5.3. BDNF in Schizophrenia

Schizophrenia, a neurodevelopmental psychiatric disorder that affects almost 1% of the general population [165] manifests with positive and negative symptoms, including psychosis, decreased expression of emotions, catatonia, and social and occupational decline [111,166]. The “mesolimbic hypothesis” is a well-established explanation for schizophrenia pathophysiology, attributing impairments in dopaminergic modulation in mesolimbic areas, particularly the striatum, to the disorder’s etiology. Altered dopamine projections onto the striatum from various brain regions contribute to its role as an integrative signaling hub [111,165].

The prefrontal cortex is also heavily implicated in schizophrenia onset, where impairments in parvalbumin neurons have been linked to schizophrenia-like phenotypes in mice [111,167,168]. Despite BDNF playing a critical role in the development and functionality of cortical parvalbumin neurons, the relationship between BDNF signaling impairments in these neurons and schizophrenia remains largely uncharacterized. Research has demonstrated that dopamine D1 receptor transmission in the dorsolateral prefrontal cortex is elevated in schizophrenia patients, regulating BDNF expression [169,170]. Furthermore, BDNF is secreted from these neurons to the striatum, where it regulates the expression of the dopamine receptor D3, which is overexpressed in schizophrenia patients [110,171,172].

BDNF haploinsufficiency causes schizophrenia-like phenotypes, which can be rescued under rich-environment conditions that promote BDNF expression and secretion in the brain [173], establishing a clear relationship between BDNF function impairments and the development of schizophrenia symptoms, influenced by environmental conditions.

Consequently, BDNF has been extensively studied in both a therapeutic target and a biomarker for schizophrenia [165,174]. Polymorphisms in BDNF and altered post-mortem BDNF brain levels have been identified in schizophrenia patients [174], and animal models have shown both decreased and increased BDNF levels in different brain regions [175,176]. However, using serum or plasma levels of BDNF as a biomarker presents challenges due to the release of BDNF from human blood platelets [177,178], leading to varied results in studies analyzing BDNF levels in schizophrenia patients [165,179].

## 6. Concluding Remarks

Neurotrophins and their receptors play a critical role in neural development and function. They exhibit distinct temporal and spatial expression patterns that not only are different at the species level, but also during their developmental stages and in brain areas.

Among the neurotrophin family, BDNF has emerged as a key element preserving neuronal health. The fine tuning of BDNF expression and its protein distribution constitutes a “spatial code” that stablishes the background for its actions through TrkB interaction. BDNF/TrkB signaling is key for GABAergic neuron development, maintenance, and function, being crucial for maintaining the excitatory/inhibitory equilibrium in the brain. 

Imbalances in both BDNF expression and its signaling cause significant implications of this neurotrophin in various diseases. Different studies support the notion that impairments in the GABAergic system driven directly or indirectly by BDNF may play an important role in the pathophysiology of neurodevelopmental disorders like autism spectrum disorder, Rett syndrome, and schizophrenia, as mentioned. 

Despite the fact that distinct effects of BDNF signaling in the brain are well characterized, the underlying mechanisms intrinsic for each neuronal population that modulate the effects of this signaling remain quite unknown. Further research is needed to reveal the complex but precise molecular interactions that drive BDNF expression and activity, possibly leading to the discovery of novel therapeutic targets that overcome the challenge of neurodevelopmental diseases.

## Figures and Tables

**Figure 3 ijms-25-08312-f003:**
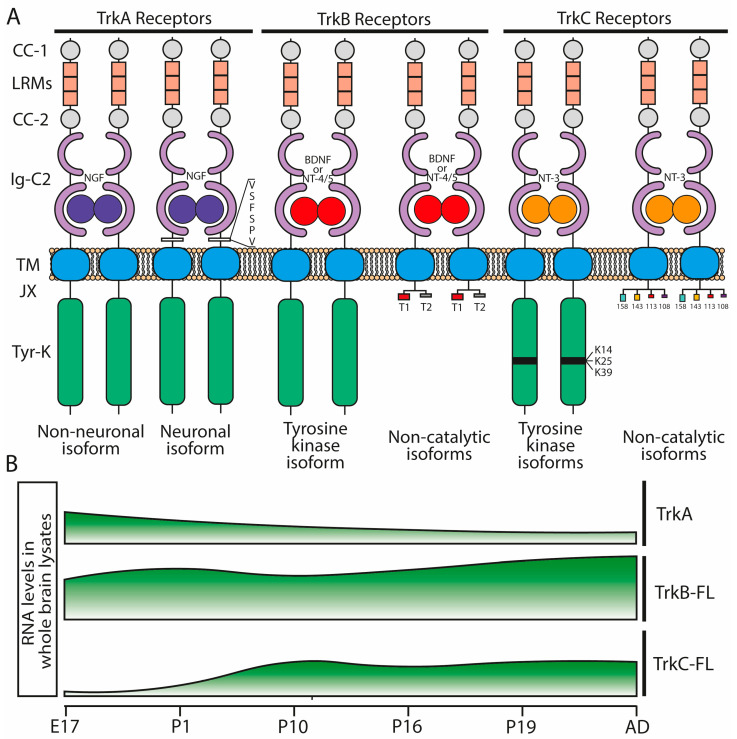
Trk neurotrophin receptors. (**A**): Structure and isoforms of the Trk receptors. Trk receptors consist of cysteine clusters (CC-1 and CC-2), leucine-rich motifs (LRMs), immunoglobulin-like C2-type motifs (Ig-C2), a transmembrane domain (TM), a juxtamembrane domain (JX), and a catalytic tyrosine-kinase domain (Tyr-K). TrkA isoforms bind NGF and differ on the six amino acid residues (VSFSPV) present only in the neuronal-specific TrkA receptor. TrkB isoforms bind BDNF and NT-4/5 and include the catalytic tyrosine kinase full-length isoforms and the non-catalytic truncated ones (T1 and T2). TrkC has three catalytic tyrosine kinase isoforms (K14, K25, and K39) and four non-catalytic truncated isoforms (TrkC^TK-158^, TrkC^TK-143^, TrkC^TK-113^, and TrkC^TK-108^) that bind NT-3. Adapted from reference [34]. (**B**): Schematic representation of Trk receptor catalytic isoform expression along rat development. Data acquired from northern blot assays from reference [41].

**Figure 4 ijms-25-08312-f004:**
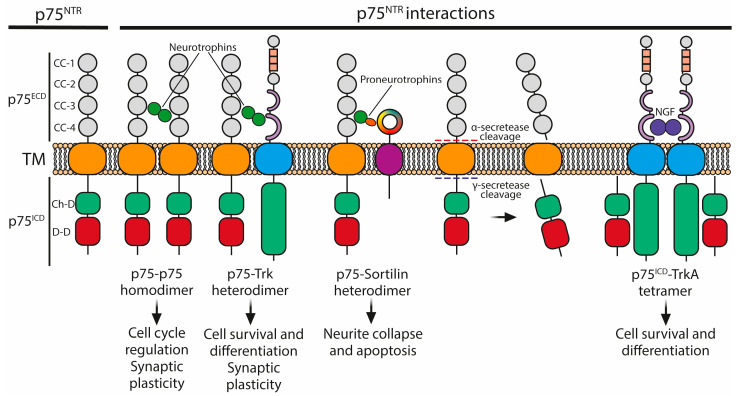
p75^NTR^ structure and interaction with other receptors. p75^NTR^ consists of an extracellular domain (p75^ECD^), consisting of four cysteine clusters (CCs), a transmembrane domain (TM), and an intracellular domain (p75^ICD^), formed by a chopper domain (Ch-D) and a death domain (D-D). p75^NTR^ can form homodimers or heterodimers (with Trk and sortilin) to exert different functions. Also, p75^NTR^ can be cleaved by the α- and γ-secretases to separate its extracellular and intracellular domains. p75^ICD^ can then bind to TrkA receptor dimers to modulate their affinity for NGF, forming a tetramer complex. Adapted from references [39,40].

**Figure 5 ijms-25-08312-f005:**
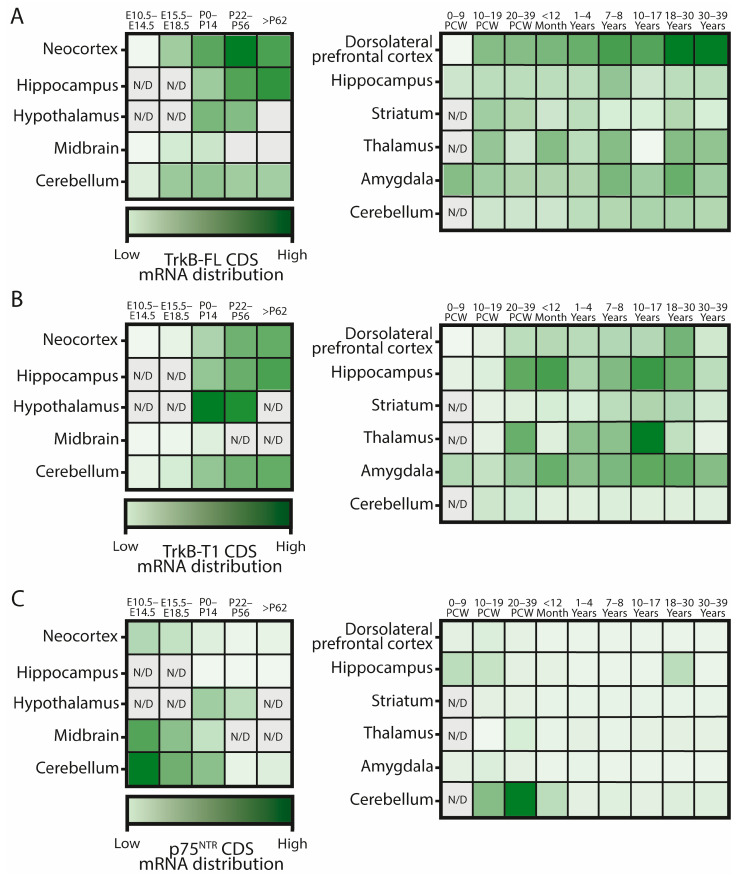
*TrkB-FL* (**A**), *TrkB-T1* (**B**), and *p75^NTR^* (**C**) mRNA distribution and relative levels in murine (left) and human (right) brains during development, according to RNAseq data from reference [17]. Dimmer colors mark higher mRNA levels, relativized to the maximum expressing point for each graph. E: mouse embryonic stage; P: mouse postnatal day; PCW: human postcoital week. N/D: data not available.

## Data Availability

Data were obtained from and are available in the references indicated in each figure.

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
