# Peer review of "Neurotrophins and Their Receptors: BDNF’s Role in GABAergic Neurodevelopment and Disease"

_ijms, 2024, doi:10.3390/ijms25158312_

Round 1

Reviewer 1 Report

Comments and Suggestions for Authors

Please find my comments and suggestions below.

Line 14 “expression pattern and roles of neurotrophins and their receptors in both the developing and adult brain” – in what species?  When referring to any findings from any publication please always indicate species you are talking about.

 ‘Disruptions in BDNF levels” – and perhaps an impact of particular single nucleotide polymorphisms.

When talking about expression of BDNF one cannot just say “BDNF transcripts” as there are many different transcripts of BDNF, perhaps with cell-specific patterns of transcription, specific sub-cellular patterns of localization (mentioned  by the authors in line 134), different mechanisms of transcription regulation and different functions.

Regarding the Figure 1. What types of transcripts? What types of BDNF protein (precursor BDNF, processed BDNF, BDNF within the cells or secreted BDNF? BDNF secreted within endosomes? BDNF secreted as a precursor, and so on and so forth)?

The first letter of the gene symbols for mice have to be capitalized, all other letters have to be lower case letters.  

Line 158 “Surprisingly, Trk receptors can also be activated in the absence of neurotrophins by transactivation mechanisms”. Please elaborate on this subject

Is there receptor-independent (of TrK-independent) BDNF-signaling?

When speaking about “low” and “high” levels of BDNF, what are the exact numbers (or a range of concentrations)?

Is there a change of BDNF expression in response to activation of neurons? How does synaptic potentiation affect BDNF expression?

Is there anything we can gain from induced pluripotent cells (iPSC) and neural progenitor cells (NPC)-based models of human diseases, in terms of the role of BDNF, disease-specific features of its transcription/translation/processing/degradation?  Speaking about BDNF degradation, what is a half-life of BDNF protein and mRNAs?

Line 348. “Specific populations of cortical neurons” – which populations?

Based on all my comments above, I conclude that the review is far from being comprehensive and complete.  It superficially summarizes some of the data, without going much into details. It also lacks a structure, because data from different animal studies (monkey, rodents, etc) are mixed/combined.  

Author Response

1.- Line 14 “expression pattern and roles of neurotrophins and their receptors in both the developing and adult brain” – in what species?  When referring to any findings from any publication please always indicate species you are talking about.

We sincerely thank the reviewer for pointing this out. We agree with this comment. Therefore, we have proceeded to indicate the species in the abstract.

2.- ‘Disruptions in BDNF levels” – and perhaps an impact of particular single nucleotide polymorphisms.

The reviewer has correctly highlighted that the expression of BDNF levels can be impacted by single nucleotide polymorphisms (SNPs), such as the Val66Met polymorphism, which has been extensively documented in the existing scientific literature. We have duly acknowledged this fact in the abstract (line 17). However, in the interest of maintaining a succinct and focused review on the overall role of BDNF, we have made the decision not to include a separate section specifically dedicated to discussing the Val66Met and other polymorphisms described in the literature. Our rationale for this decision stems from the fact that the corresponding author, Dr. Deogracias, recently contributed to an extensive review that comprehensively addressed the mechanisms governing BDNF expression and secretion, including an examination of the Val66Met polymorphism. We are hopeful that reviewer 1 will concur with our reasoning behind excluding this additional section.

3.- When talking about expression of BDNF one cannot just say “BDNF transcripts” as there are many different transcripts of BDNF, perhaps with cell-specific patterns of transcription, specific sub-cellular patterns of localization (mentioned by the authors in line 134), different mechanisms of transcription regulation and different functions.

Considering this comment, we have revised the introductory paragraph of section 2.3 to incorporate additional references that elucidate the presence of diverse BDNF transcripts and their respective functional roles.

4.- Regarding the Figure 1. What types of transcripts? What types of BDNF protein (precursor BDNF, processed BDNF, BDNF within the cells or secreted BDNF? BDNF secreted within endosomes? BDNF secreted as a precursor, and so on and so forth)?

Consequently, we have revised the figure legend of Figure 1. Upon careful examination of all the references and data utilized in constructing this figure, we have specified that the RNA data pertain to the coding sequence (CDS) of each neurotrophin. Furthermore, it is important to note that the antibodies employed by the referred publications did not differentiate between precursor and mature neurotrophins.

5.- The first letter of the gene symbols for mice have to be capitalized, all other letters have to be lower case letters.  

Given that the data incorporated in this review encompass multiple species including mouse, rat, monkey, and human, our initial decision was made to employ uppercase letters for gene symbols and proteins. However, in response to the reviewer's correct observation and with the intention of avoiding any confusion for readers, we have adhered to the international criteria as recommended by the reviewer.

6.- Line 158 “Surprisingly, Trk receptors can also be activated in the absence of neurotrophins by transactivation mechanisms”. Please elaborate on this subject

We discussed and elaborated on this subject on the first edition of this manuscript. Therefore, we have not made any changes on this subject apart of indicating the word transactivated on the section in order to make it clear.

7.- Is there receptor-independent (of TrK-independent) BDNF-signalling?

Following a thorough examination of the available literature, we have not encountered any references suggesting that BDNF possesses the capability to initiate signalling pathways in the absence of any specific TrkB isoforms or the p75 neurotrophic receptor. We kindly request the reviewer's assistance in identifying any peer-reviewed scientific publications that we could potentially incorporate into our manuscript to facilitate a discussion on this subject matter if he/she still consider it is really necessary.

8.- When speaking about “low” and “high” levels of BDNF, what are the exact numbers (or a range of concentrations)?

Despite we knowledge of the documented low levels of BDNF in the brain, quantified in terms of nanograms (ng) of the protein per gram (g) of wet brain tissue, as reported in various studies in the literature (see for example Rauskolb et al., 2010; Deogracias et al., 2012), the data presented in Figure 2b were acquired from the study conducted by Tonis Timmusk lab (Esvald et al., 2023). These data encompass densitometric quantifications of western blots from different brain regions of BALB/c and C57BL/6J mice throughout their postnatal development. As Esvald and collaborators did not provide information regarding the concentration or quantity of BDNF in their samples in the indicated publication, we have applied the same criteria utilized by the authors for the interpretation and analysis of these data.

9.- Is there a change of BDNF expression in response to activation of neurons? How does synaptic potentiation affect BDNF expression?

In our manuscript, we focus on delineating the expression levels of BDNF during brain development. Although we acknowledge that various stimuli, including neuronal activity, and synaptic potentiation can modulate BDNF expression, we have intentionally excluded an in-depth discussion on these topics as they fall outside the scope of our intended review. We hope the reviewer could understand and agree with our decision.

10.- Is there anything we can gain from induced pluripotent cells (iPSC) and neural progenitor cells (NPC)-based models of human diseases, in terms of the role of BDNF, disease-specific features of its transcription/translation/processing/degradation?  Speaking about BDNF degradation, what is a half-life of BDNF protein and mRNAs?

We express our sincere appreciation to the reviewer for bringing up this point. The utilization of certain models is crucial in providing valuable insights into the role of BDNF. It is well recognized that iPSCs and NPCs have been extensively used by numerous authors to investigate human diseases and the involvement of BDNF in regulating various aspects of neuronal activity, such as transcription, translation, processing, degradation, neuronal survival, and epigenetic regulations. However, during the initial planning of our manuscript, we made a deliberate decision to concentrate on animal models and data from human patients to avoid potential misinterpretations that could arise from in vitro findings. Consequently, we deemed it unnecessary to include a review of these specific models in our present work.

With regards to BDNF degradation, specifically in terms of protein and mRNA half-life, our thorough literature search revealed that circulating BDNF possesses a half-life of less than 10 minutes, as reported by Sakane and Pardridge in 1997. However, despite conducting an extensive bibliographic investigation, we were unable to find any information regarding the in vivo half-life of BDNF in different species. Although we speculate that it might be similar to NGF based on the findings of Saltzman et al. in 1996, we opted not to include this detail in our review. We kindly request the reviewer to indicate the significance of incorporating this information into the text if deemed necessary.

11.- Line 348. “Specific populations of cortical neurons” – which populations?

We extend our appreciation to the reviewer for raising this valuable comment, and we concur that it is crucial to offer additional elucidation on this subject matter. As a response to this feedback, we have incorporated a more comprehensive explanation regarding the secretion of BDNF in the corticostriatal pathways. Furthermore, we have addressed the functional implications stemming from the anatomical division of projections from different cortices to the striatum. Through this addition, our intention is to augment the comprehension of the mechanisms underlying BDNF secretion within the corticostriatal pathways.

12.- Based on all my comments above, I conclude that the review is far from being comprehensive and complete.  It superficially summarizes some of the data, without going much into details. It also lacks a structure, because data from different animal studies (monkey, rodents, etc) are mixed/combined.  

We express our gratitude to the reviewer for his/her valuable feedback, which has contributed to enhancing the comprehensiveness, completeness, and clarity of our manuscript's structure.

Reviewer 2 Report

Comments and Suggestions for Authors

In this manuscript, the Authors summarized different aspects of neutrophins and of their receptors, analysing their distribution, synthesis and structure in the central nervous system. In addition, they focused on BDNF, attemptig to summarize the current knowledge about the role of this neurotrophin and its receptor in GABAergic neurons and to explore the role of BDNF in neurodevelopmental disorders with impairments in GABAergic neurons.

The manuscript is well written and distributed, the first part is very interesting and the figures are well made, showing clearly in the same panel the distribution and the level of expression of the different neutrophins at different stages of neurodevelopment. However, the second part of the paper on the role of BDNF on neurodevelopmental disorders in less interesting and a little confusing, it should be reviewed.

Major points

The authors want to explain the role of BDNF in ASD, from data in 5.1 paragraph, the involvement of the neurotrophin in ASD is not clear: BDNF appears only a consequence of the pathology, if BDNF can have an active role in ASD should be better explained.

The authors should consider to merge 5.1 and 5.2 paragraphs based on the following considerations:

- Rett syndrome is classified as an autism spectrum disorder,

- There is some overlap between the paragraphs (line 405-407, this sentence should be included in rett syndrome paragraph)

In general, it is not clear the aim of the authors in the last part of the paper: if they want to explain the role of BDNF in neurodevelopmental disorders characterized by GABA transmission impairment or if they want to analyse if BDNF alterations can have a role in the impairment of GABA neurons in these pathologies. In particular this second aspect is not clear and should be clarified.

Minor point

Line 65: based on figure 1B “ and in the cerebellum” should be added after the word “hippocampus”.

Author Response

In this manuscript, the Authors summarized different aspects of neutrophins and of their receptors, analysing their distribution, synthesis and structure in the central nervous system. In addition, they focused on BDNF, attemptig to summarize the current knowledge about the role of this neurotrophin and its receptor in GABAergic neurons and to explore the role of BDNF in neurodevelopmental disorders with impairments in GABAergic neurons.

The manuscript is well written and distributed, the first part is very interesting, and the figures are well made, showing clearly in the same panel the distribution and the level of expression of the different neutrophins at different stages of neurodevelopment. However, the second part of the paper on the role of BDNF on neurodevelopmental disorders in less interesting and a little confusing, it should be reviewed.

We extend our appreciation to the reviewer for their insightful comments regarding the initial section of our manuscript, highlighting its intriguing nature, as well as acknowledging the clarity and quality of the figures presented. Furthermore, we would like to express our gratitude for the reviewer's diligent efforts in providing suggestions to enhance the structural coherence of the second part of our manuscript.

Major points

1.- The authors want to explain the role of BDNF in ASD, from data in 5.1 paragraph, the involvement of the neurotrophin in ASD is not clear: BDNF appears only a consequence of the pathology, if BDNF can have an active role in ASD should be better explained.

We value the reviewer's comment and would like to emphasize that in the initial version of our manuscript, we concluded that "The association between BDNF alterations and ASD remains complex, encompassing both causative and consequential factors." Furthermore, we concluded the section by acknowledging the intricacies of comprehending the role of BDNF in ASD, given the multifaceted nature of this spectrum disorder.

The authors should consider to merge 5.1 and 5.2 paragraphs based on the following considerations:

- Rett syndrome is classified as an autism spectrum disorder,

- There is some overlap between the paragraphs (line 405-407, this sentence should be included in rett syndrome paragraph).

We thank the reviewer for this comment. Despite Rett Syndrome has been considered for many scientists as part of the ASDs, since the last edition of “The Diagnostic and Statistical Manual of Mental Disorders, Fifth Edition” (DSM-V), Rett Syndrome is classified as a neurodevelopmental disorder but not as an Autism Spectrum Disorder. Therefore, we have maintained the original structure of the manuscript as it was originally written following the classification indicated by the DSM-V.

In general, it is not clear the aim of the authors in the last part of the paper: if they want to explain the role of BDNF in neurodevelopmental disorders characterized by GABA transmission impairment or if they want to analyse if BDNF alterations can have a role in the impairment of GABA neurons in these pathologies. In particular this second aspect is not clear and should be clarified.

To provide clarity on the matter, we have removed the words "role of" from all section 5 and subsection titles. We consider that this adjustment allows us to specifically focus on discussing the alterations in BDNF expression and its signalling in ASD, Rett Syndrome, and schizophrenia, where GABAergic neuronal impairments have been observed.

Minor point

Line 65: based on figure 1B “ and in the cerebellum” should be added after the word “hippocampus”.

In response to the reviewer's comment, we have made the decision to refrain from including "and in the cerebellum" in this particular line. This choice is based on the fact that we are specifically discussing early maturating areas, while the cerebellum is recognized as a late maturating region according to existing literature. Additionally, Figure 1B indicates that BDNF expression in the cerebellum is lower compared to the other areas mentioned in the original statement at line 65.

Round 2

Reviewer 1 Report

Comments and Suggestions for Authors

The review has improved significantly and I am very glad that the authors found some of my comments and suggestions instrumental for further improving the text.

In line 115, to avoid confusion with gene name nomenclature, I suggest the authors to write something along the lines "In both mice and human the gene encoding BDNF contains several 5′-non-coding...."

Reviewer 2 Report

Comments and Suggestions for Authors

No suggestions. All issues are solved